# Review on Pediatric Malignant Focal Liver Lesions with Imaging Evaluation: Part II

**DOI:** 10.3390/diagnostics13243659

**Published:** 2023-12-14

**Authors:** Yi Dong, Andrius Cekuolis, Dagmar Schreiber-Dietrich, Rasa Augustiniene, Simone Schwarz, Kathleen Möller, Nasenien Nourkami-Tutdibi, Sheng Chen, Jia-Ying Cao, Yun-Lin Huang, Ying Wang, Heike Taut, Lara Grevelding, Christoph F. Dietrich

**Affiliations:** 1Department of Ultrasound, Xinhua Hospital, School of Medicine, Shanghai Jiao Tong University, Shanghai 200092, China; drdaisydong@hotmail.com (Y.D.); 21211210033@m.fudan.edu.cn (S.C.); califfa@126.com (J.-Y.C.); 22111210067@m.fudan.edu.cn (Y.-L.H.); yunxi2009@126.com (Y.W.); 2Ultrasound Section, Department of Pediatric Radiology, Radiology and Nuclear Medicine Centre, Vilnius University Hospital Santaros Klinikos, 08661 Vilnius, Lithuania; andrius.cekuolis@gmail.com (A.C.); augustiniene.rasa@gmail.com (R.A.); 3Localinomed and Pediatric Department, Kliniken Salem, 3013 Bern, Switzerland; dietrich.dagmar@googlemail.com; 4Department of Neonatology and Pediatric Intensive Care Medicine, Sana Kliniken Duisburg GmbH, 47055 Duisburg, Germany; simone.schwarz@sana.de; 5Medical Department I/Gastroenterology, SANA Hospital Lichtenberg, 10365 Berlin, Germany; k.moeller@live.de; 6Saarland University Medical Center, Hospital of General Pediatrics and Neonatology, 66421 Homburg, Germany; nasenien.nourkami@uks.eu; 7Children’s Hospital, Universitätsklinikum Dresden, Technische Universität Dresden, 01069 Dresden, Germany; heike.taut@uniklinikum-dresden.de; 8Department of Pediatrics, Division of Pneumology, Allergology, Infectious Diseases and Gastroenterology, University Hospital Frankfurt, Goethe University, 60323 Frankfurt, Germany; 9Department Allgemeine Innere Medizin (DAIM), Kliniken Hirslanden Beau Site, Salem und Permanence, 3013 Bern, Switzerland

**Keywords:** pediatric, malignant, focal liver lesions (FLLs), diagnosis, imaging, contrast-enhanced ultrasound (CEUS)

## Abstract

Malignant focal liver lesions (FLLs) represent various kinds of epithelial and mesenchymal tumors. In pediatric patients, the understanding of pediatric liver diseases and associated imaging manifestations is essential for making accurate diagnosis and differential diagnosis. This paper will discuss the latest knowledge of the common pediatric malignant FLLs, including undifferentiated embryonal sarcoma, rhabdomyosarcoma, epithelioid hemangioendothelioma, angiosarcoma, and malignant rhabdoid tumor. Medical imaging features are not only helpful for clinical diagnosis, but can also be useful in the evaluation and follow-up of pre- and post-treatment. The future perspectives of contrast-enhanced ultrasound (CEUS) enhancement patterns of FLLs in pediatric patients are also mentioned.

## 1. Introduction

Primary hepatic tumors, which can be benign, malignant, or indeterminate, comprise only 0.5–2.0% of all pediatric tumors [1]. The reported incidence is 0.4 to 1.9 per year per million children, varying with entity and age. In pediatric patients, malignant focal liver lesions are the third most common solid intra-abdominal malignancy [2,3].

Among all primary malignant hepatic tumors in pediatric patients, hepatoblastoma is the most common one. It was reported that most hepatoblastomas occurred in the first two years of life and we refer to the first part of this paper [4]. Very sparsely diagnosed and described in children are mesenchymal malignant tumors of the liver, including epithelioid hemangioendothelioma, undifferentiated embryonal sarcoma, rhabdomyosarcoma, and angiosarcoma [1,2]. Other rare malignancies are malignant rhabdoid tumor and germ cell tumor [5]. Liver metastases can occur as a result of other childhood tumors such as neuroblastoma, Wilms’ tumor, Ewing sarcoma (Figure 1), or lymphoma [6]. In nephroblastoma (Wilms’ tumor) and neuroblastoma, tumor plugs within the inferior vena cava can be found reaching up to the right atrium in some cases [7]. Various medical imaging methods are commonly used clinically in pediatric patients [8]. Imaging features and improved imaging modalities help to better understand pediatric liver tumors and to establish accurate diagnosis [6]. Generally, ultrasound is used as an initial screening technique to detect the lesion and rule out any other abnormalities, followed by computed tomography (CT) and/or magnetic resonance imaging (MRI) to complete the characterization and staging of tumors [9]. The use of CT or MRI before surgery is mandatory. All mentioned imaging techniques have their advantages and disadvantages.

Because hepatic sarcomas are extremely rare, only small case studies and case reports have been published. Hepatic sarcomas, like other FLLs, can be classified into different types according to their growth pattern and degenerative histological features: a large solitary mass, a mixed pattern with a dominant mass associated with satellite nodules, multiple nodules, and the rare form of diffuse infiltrating micronodular tumor and metastases [10]. Pathologically, angiosarcomas are characterized by areas of hemorrhage and necrosis [11]; therefore, imaging features of hepatic angiosarcoma show typically bizarre heterogenous lesions, which might be sometimes misdiagnosed as atypical gigantic hemangioma with non-enhancing areas [12]. Coarse nodular enhancement with or without centripetal filling, combined with the presence of a rim and reticular “septal-like” chaotic arterial enhancement and washout, should prompt further diagnostic workup including a needle biopsy. It is of importance that washout might be late, sometimes after two and up to five minutes.

The majority of FLLs could be well characterized on imaging diagnosis, while histopathological diagnosis via biopsy or surgery resection is required for confirmation and/or further examination to plan therapy regimes [1,4]. However, biopsies, whether Tru-Cut or open, contain a significant rate of complications [13].

The aim of the current review is to summarize imaging features of malignant FLLs in pediatric patients. Emphasis will be laid on the knowledge and treatment strategies of FLLs based on various imaging features.

## 2. Undifferentiated Embryonal Sarcoma of the Liver (UESL)

### 2.1. Epidemiology, Clinical Features, and Pathological Features

Undifferentiated embryonal sarcoma of the liver (UESL) is an aggressive malignant FLL most seen in the pediatric patients. It accounts for 9–15% of pediatric liver cancers and is the most common sarcoma as well as the third most common liver malignancy seen in pediatric patients [14]. From 1978 to 2014, 198 children with UESL were reported in 23 case series [15]. Most cases of UESL are diagnosed in children aged 6–10 years, with a slight male predominance [16,17].

Children with UESL usually present a palpable abdominal mass and abdominal pain [18]. Other non-specific systemic clinical manifestations include fever, nausea, anorexia, vomiting, diarrhea, and weight loss [19]. Fever is usually associated with hemorrhage and necrosis within the tumor [20]. Acute presentation secondary to its rupture/wall dehiscence due to its rapid growth has also been reported [21]. The serum AFP levels are usually normal, making detection and monitoring with blood test obsolete in this type of liver tumors. Metastases occur in up to 15% of pediatric patients, most commonly involving the peritoneum, lungs, and pleura [22].

UESL is often a large (usually larger than 10 cm at diagnosis) and predominantly solid tumor of mesenchymal origin, located in the right lobe of the liver [6]. Macroscopically, most USELs demonstrate a well-circumscribed single lesion with a fibrous pseudo-capsule. UESL has a cut surface which reveals a heterogenicity of gray-white firm tissue, foci of hemorrhage and necrosis, and cystic gelatinous areas [23]. Microscopically, on histologic examination, UESL is predominantly composed of sarcomatous stellate or spindle-shaped cells that are alternately compactly and loosely arranged in an abundant myxoid stroma [24]. A pseudo-capsule separates the tumor mass from the adjacent liver parenchyma. Meanwhile, cords and clusters of hepatocytes are also commonly observed within the pseudo-capsule and at the peripheral margins of the lesion [25]. Mitoses are commonly identified throughout the tumor.

### 2.2. Imaging Features

In the literature, the imaging characteristics of UESL have been described, reflecting its solid, cystic, and mucoid composition. Also, it appears predominantly solid on conventional B mode ultrasound, and shows cystic features on CT and MRI due to its high-water content of the prominent myxoid stroma and areas of hemorrhage [26].

Ultrasound: On ultrasound, UESL appears as a heterogeneous tumor, mostly hyperechoic with small anechoic cyst-like spaces, which are associated with hemorrhage, necrosis, and cystic degeneration [27]. Moreover, a minority of lesions that were predominantly anechoic and composed of fluid-filled spaces separated by septa, simulating benign tumors, have also been reported [28].

CEUS: CEUS performed on patients with UESL has been rarely reported. UESL might show minimal hypoenhancement due to its complex components, which make it difficult to distinguish from liver abscess. However, biopsy can be conducted under the guidance of CEUS to improve accuracy and ensure the acquisition of positive samples (Figure 2).

CT: CT reveals a solitary, well-defined, and predominantly hypodense cystic mass, while solid nodules and septations are also present [29]. Internal septations, the papillary portion, and the periphery of the tumor region may show slight enhancement on CECT [30]. The enhancing peripheral rim on delayed images corresponds to the pseudo-capsule seen at the macroscopic inspection of the tumor. Central foci of high attenuation representing acute hemorrhage could be observed [26]. Internal calcifications and serpiginous vessels within the tumors can be occasionally seen on CECT [31]. Gabor et al. [28] conducted a retrospective study of 15 children diagnosed with UESL, and serpiginous vessels that might be of an arterial origin were observed in 10 of the cases during the early enhanced acquisition. They even appeared when the disease recurred 3.5 years later and were not present at initial diagnosis solely in one case among those with recurrence. Serpiginous vessels may be an important finding in diagnosing UESL when a hypodense cystic-like tumor appearance is observed on CT, and might potentially serve as an imaging biomarker.

MRI: On MRI, UESL usually shows as a well-defined mass, predominantly hypointense on T1-weighted images and hyperintense on T2 weighted images [30]. The hypointense rim appearance on T1- and T2-weighted images represents the fibrous pseudo-capsule [32]. Areas of hemorrhage present heterogeneously hyperintense on T1-weighted images and hypointense on T2-weighted images and fluid-fluid levels, which are better seen with MRI than with CT [33]. In addition, the internal details of tumors such as solid components and septa showing progressive enhancement on delayed images are also better revealed on MRI [4]. After injection of contrast agents, a heterogeneous hyperenhancement of the tumor could be observed. MRI is better than CT scan evaluating the involvement of the venous structures, in determining the tumor resectability, as well as in detecting the potentially affiliated biliary tree and lymph nodes [6].

### 2.3. Management and Prognosis

The use of multimodal treatment including chemotherapy, surgical resection, and transplantation makes UESL possibly curable [22]. Most UESLs respond to neoadjuvant chemotherapy, which renders some surgically unresectable tumors amenable to resection and becomes the mainstay of cure. Liver transplantation is suggested for an unresectable USEL localized to the liver [34].

Early diagnosis is critical to improve long-term survival. The prognosis of UESL has improved with a 5-year overall survival of 80–100% in recent years, with a multi-disciplinary and -modal approach [4]. Shi et al. [22] identified a total of 103 patients under 18 years with UESL, with a 5-year overall survival of 86%. In children who underwent a combined therapy of chemotherapy and surgical resection, the 5-year overall survival was 92%. Orthotopic liver transplantation was performed in 10 pediatric patients resulted in an overall survival rate of more than 5 years within the observation interval. Multivariate analysis revealed that a tumor size ≥ 15 cm (*p* = 0.02) and combined therapy (*p* < 0.01) were the only two independent prognostic factors.

## 3. Biliary Rhabdomyosarcoma (BRMS)

### 3.1. Epidemiology, Clinical Features, and Pathological Features

Rhabdomyosarcoma is the most common soft-tissue sarcoma in pediatric patients. Biliary rhabdomyosarcoma is the botryoid subtype of embryonal rhabdomyosarcoma with localization in the epithelial lined biliary tract. Therefore, in the literature, these tumors are also referred to as embryonal rhabdomyosarcoma of the liver [35]. It is a rare but highly aggressive tumor, accounting for 1% of all pediatric liver tumors [4]. BRMS occurs almost exclusively in children, and is mostly diagnosed under the age of 5 years with a slight male predominance [36]. The median age at diagnosis is 3 years.

BRMS can develop from both the intra- and extrahepatic biliary tract, with the CBD being the most commonly primary site [37]. Jaundice is the most prevalent symptom in BRMS, and BRMS is the most common malignant cause of obstructive jaundice in pediatric patients [38]. Children may also present with abdominal pain, abdominal distension, fever, nausea, vomiting, and hepatomegaly [39]. Patients may have elevated levels of conjugated bilirubin and alkaline phosphatase and normal serum AFP levels. Approximately 30% of cases present with metastases at the time of initial diagnosis [40].

The tumors generally display a polypoid or botryoid small cystic growth pattern within the wall of the bile duct and gradually extend into the lumen. The tumor mostly originates in extrahepatic bile ducts but may grow into intrahepatic biliary ducts and finally invade the liver consecutively [4]. Histologically, BRMS presents as an embryonal subtype with plump and round or spindle-shaped rhabdomyoblasts, loosely arranged in a myxoid stroma beneath the cambium layer, possibly demonstrating characteristic cross-striations [6].

### 3.2. Imaging Features

Ultrasound: Ultrasound is frequently applied in the initial stages of the diagnostic process, especially in individuals with obstructive jaundice. Biliary dilation is often demonstrated on ultrasound, and BRMS typically shows as a single heterogeneous hypoechoic or isoechoic lesion, or multiple hypoechoic lesions separated by septa within the lumen, commonly with an associated displacement of the portal vein without intraluminal thrombus [41]. Cystic areas possibly caused by tumor necrosis can be observed in larger masses. Roebuck et al. [42] reported one case of BRMS showing mixed echogenicity with numerous pathological arteries with low resistance index on color Doppler imaging. In one case reported by Chavhan et al., in the dilated bile duct, an iso- to hypoechoic soft tissue lesion with vascularity was visualized on ultrasound [4].

CEUS: So far, there are no CEUS experiences in the literature on BRMS in pediatric patients.

CT: BRMS can be variable in density on CT images, showing as a intraductal homo- or heterogeneous hypo- or hyperdense mass, which may have fluid-attenuation components, with or without biliary dilatation [43]. In addition to typical features, Kitagawa et al. [44] described a 2-year-old boy histologically confirmed with BRMS by biopsy, showing a completely multilocular cystic form probably due to tumor necrosis. The enhancement patterns are also highly variable and may show an intense heterogeneous, incomplete globular, mild, or even nonenhancement pattern [42].

MRI: On MRI, BRMS typically appears as a predominantly fluid-intensity mass which shows as hypointense on T1-weighted images, and moderately or markedly hyperintense on T2-weighted images [45]. CEMRI demonstrates heterogeneous enhancement within the solid components of the mass and intraductal material within the biliary tree, which may allow to distinguish BRMS from a choledochal cyst filled with sludge. The latter can present with similar imaging features of central hemorrhage and necrosis. Magnetic resonance cholangiopancreatography (MRCP) may illustrate a dilated irregular common bile duct with large filling defects, a distension of the gallbladder, and dilated intrahepatic bile ducts and pancreatic duct [46].

Other imaging techniques: Percutaneous transhepatic cholangiography (PTC) or endoscopic retrograde cholangiography (ERCP) frequently reveal the dilated common bile duct with a large filling defect, with or without an obstruction of the extrahepatic bile ducts [47]. PTC offers advantages in patients with coagulopathy, while ERCP is better in defining the degree of intraductal extension and in patients with obstructive jaundice for adequate biliary drainage [42]. FDG-PET/CT or PET/MRI along with chest CT are recommended for the detection and evaluation of loco-regional and distant metastatic diseases [48].

### 3.3. Management and Prognosis

BRMS is generally managed using a multi-disciplinary and -modal treatment approach, including surgery, radiation, and chemotherapy, based on specific risk stratifications. Complete surgical resection is only feasible in 20–40% of all cases. Aggressive surgery may be unwarranted according to previous reports, and the outcomes remain relatively good despite residual disease after surgery [39]. It has been reported that the surgical resection of the residual tumor after neoadjuvant chemotherapy shows favorable outcomes [47]. ERCP may be required to relieve biliary obstruction [49]. In addition, the combined treatment of liver transplantation and chemotherapy has been proved to be effective for unresectable BRMS without extrahepatic metastases in children, according to previous case reports [50,51,52].

The improved prognosis of BRMS showed an estimated 5-year survival up to 66% [53]. Positive predictive factors for survival include age ≤ 10 years and botryoid tumor histology [54]. Metastases occur at diagnosis in ≥30% of all cases with extrahepatic diseases and show a poor prognosis [6].

## 4. Hepatic Epithelioid Hemangioendothelioma (HEHE)

### 4.1. Epidemiology, Clinical Features, and Pathological Features

Epithelioid hemangioendothelioma is an extraordinarily rare sarcoma of vascular endothelial origin. It has an estimated prevalence of less than one in one million, which can arise at multiple and different anatomic sites, most frequently in the liver, lungs, bones, and soft tissue [55]. Hepatic epithelioid hemangioendothelioma (HEHE) is an intermediate malignant potential tumor in-between a benign IHH and a highly aggressive angiosarcoma [6]. The median age of children with HEHE at diagnosis is 12 years, with a female predominance [56]. Approximately 25% of patients with HEHE are asymptomatic, while nonspecific clinical symptoms and signs, including right upper quadrant pain, weight loss, hepatomegaly, and a palpable mass, are present in symptomatic patients [57]. HEHE may also exhibit as veno-occlusive disease or Budd–Chiari syndrome in rare cases [58]. The levels of serum tumor markers, such as AFP levels, are normal [6].

Macroscopically, HEHE most often appears as firm, multifocal, ill-defined, and sometimes focally confluent nodules with infiltrative margins and peripheral congestion or hyperemia, resulting in capsular retraction. It shows a predominantly peripheral or subcapsular growth pattern, which usually involves both hepatic lobes. It may also occur as a solitary mass [59]. Microscopically, on histologic examination, the tumor consists of dendritic, epithelioid, and intermediate cells with vascular differentiation in a fibromyxoid stroma, which grow along vascular structures, infiltrate hepatic sinusoids, and disrupt hepatic plates [4]. Immunohistochemistry shows positive staining for factor VIII-related antigen, ERG, FLI-1, CD31, CD34, and D2-40, being critical for an accurate diagnosis of HEHE [60,61]. HEHE must be distinguished from the infantile hemangioendothelioma of the liver, which occurs in the newborn and is more likely to be classified as a congenital hemangioma.

### 4.2. Imaging Features

On imaging, HEHE demonstrates two main patterns of the progressive disease, including the nodular type that consists of multifocal and predominantly peripheral nodules in the early stage, and the diffuse type when enlarged nodules merge into confluent masses in the advanced stage [62].

Ultrasound: On grayscale ultrasound, HEHE manifests as unifocal, ill-defined, multifocal, or diffuse FLLs involving both lobes. The lesions are mainly hypoechoic on conventional BMUS due to central myxoid stroma, but a heterogeneous echogenicity with hypo- and hyperechoic pattern can also be observed [63]. There is no correlation between the size of the lesion and their echogenicity on BMUS [64]. Branched intralesional vessels can be detected in most lesions on color Doppler imaging. However, multiple HEHE on conventional ultrasound mostly present without any specific imaging findings, increasing its difficulty to be differentiated from other atypical multifocal liver lesions [65].

CEUS: CEUS characteristic of pediatric HEHE can refer to the experience of adult groups. During the arterial phase of CEUS, HEHE may show peripheral rim-like or heterogeneous hyperenhancement. In the portal venous phase and late phase, HEHE shows quick washout and becomes hypoenhanced with unenhanced central areas [66,67]. Typical CEUS imaging features reliably allow for effective differentiation between HEHE and other benign FLLs such as hepatic hemangioma and FNH, both showing hyperenhancement and remaining iso- or hyperenhanced in the portal venous and late phase [63] (Figure 3).

CT: On unenhanced CT, the predominantly peripheral nodules present classically hypodense in comparison to the liver parenchyma, while coalescing masses may appear as nonspecific heterogeneous appearances. Nodular or coarse calcifications with irregular spots can be detected inside the lesions in approximately 20% of cases [68]. Lesions adjacent to the capsule can show capsular retraction or flattening [64].

CECT reveals a hypodense central area, a peripheral hyperdense rim, and a more peripheral hypodense rim, described as a target-like appearance [69]. Dynamic imaging demonstrates peripheral arterial phase hyperenhancement and progressive centripetal filling in the delayed phase [4]. Alomari et al. [70] reported a series of four cases of HEHE showing a specific “lollipop sign”, which is a combination of the hypodense well-defined mass on contrast-enhanced CT images representing the candy and the histologically occluded vein representing the stick. This hallmark may improve the recognition of HEHE on cross-sectional imaging, although it might be missing in a percentage of cases.

MRI: HEHE usually appears hypointense with a more hypointense central portion on T1-weighted images and heterogeneously hyperintense with a more hyperintense central portion on T2-weighted images on MRI [6]. A typical halo appearance could be observed, consisting in a three-layered pattern with a hypo- or hyperintense core and alternating hypo- and hyperintense rims [68]. CEMRI demonstrates peripheral enhancement with gradual centripetal filling on delayed images, similar to CECT [71]. In addition, the mean apparent diffusion coefficient (ADC) value of HEHE may be relatively high in comparison with other hepatic malignancies on DWI, which might be helpful in diagnosis [69].

Other imaging techniques: FDG-PET/CT reveals that the uptake of FDG demonstrates a moderate to intense level, and is also observed in lymph nodes and extrahepatic sites [72]. Da Ines et al. [73] described a rare case of an 11-year-old boy diagnosed with HEHE, of which a PET/CT procedure was performed with a strong suspicion of coeliac nodal involvement. It was confirmed by surgery resection and histopathological results. The use of PET/CT can not only allow superior staging to CT or MRI at initial diagnosis but also evaluate the response to therapy during follow-up.

### 4.3. Management and Prognosis

Due to its rarity in pediatric patients, no standard treatment strategy for HEHE exists currently. In most patients, surgical resection is considered the best therapeutic option as the tumor does not respond effectively to chemotherapy, yet this is impossible to perform if extensive liver involvement with multiple or diffuse nodules is present [73]. Sharif et al. [74] reported a series of children with HEHE and concluded that HEHE in pediatric patients may show a more malignant behavior compared to that reported in adults, with surgery alone not being the most favorable approach in children. Liver transplantation may not be appropriate for children with unresectable HEHE, which is an accepted indication for liver transplantation in adult patients, while preoperative chemotherapy before liver transplantation should be taken into consideration in pediatric patients [75]. Meanwhile, rare cases of successful liver transplantation for pediatric HEHE have also been reported [76,77]. Guiteau et al. [78] analyzed the data of pediatric orthotopic liver transplantation for HEHE and found that the 5-year survival rate was 60.6%, with a recurrence rate of 2.8%, and a death rate of 9% for those suffering from recurrent disease.

In general, the prognosis of HEHE is better than other hepatic malignancies in pediatric patients. Meanwhile, metastatic disease at diagnosis does not imply a worse chance of survival [79]. Yet, follow-up data are scarce and long-term prognosis remains unclear.

## 5. Hepatic Angiosarcoma (HAS)

### 5.1. Epidemiology, Clinical Features, and Pathological Features

Hepatic angiosarcoma (HAS) is a rare but high-grade tumor of endothelial cells, mostly affecting elderly men but possibly occurring in children, typically in girls with a mean age of 3 years [80]. Pediatric HAS is extremely rare. It has only been described in several case reports or case series, comprising only 1–2% of pediatric liver tumors [81]. It is believed to be the malignant transformation of IHH; however, the etiology of HAS in children is still unclear [82]. The association between HAS and toxic exposures as described in adults is still uncertain among children, and only one case of pediatric HAS has been reported to be associated with arsenic exposure [83]. In addition, a few cases have shown pediatric HAS occurring on different backgrounds, such as multiple cutaneous infantile hemangiomas [84], cutaneous mixed vascular malformations [85], and dyskeratosis congenita [86].

The most common symptoms are abdominal pain and distension caused by rapid liver enlargement, accompanied by nonspecific symptoms of anorexia, nausea, vomiting, fever, anemia, and weight loss [87]. Other symptoms and signs include ascites, jaundice, dyspnea, thrombocytopenia, and liver failure [59]. Spontaneous tumor rupture can occasionally occur and lead to hemoperitoneum [88]. Previously reported complications included disseminated intravascular coagulation, consumptive coagulopathy, and congestive heart failure [80]. Metastatic disease is common at diagnosis, most commonly affecting the lungs and spleen [89].

Macroscopically, pediatric HAS may demonstrate as multiple lesions, a large solitary mass, a mixed pattern with a dominant mass associated with satellite nodules, or a rare form of diffuse infiltrating micronodular tumor [89]. Areas of hemorrhage and necrosis due to the invasion of hepatic venules and portal vein branches may be observed on cut surface [90]. On microscopy, the tumor is characterized by slightly eosinophilic, spindle or pleomorphic cells that form vascular channels, ranging from excessively dilated sinusoidal or cavernous spaces to slit-like freely anastomosing vascular channels [6]. On immunohistochemistry analysis, the neoplastic cells are positive for several endothelial markers confirming the vascular derivation of the tumor, including FLI-1, ERG, CD31, CD34, and factor VIII-related antigen [59].

### 5.2. Imaging Features

Ultrasound: HAS demonstrates heterogeneous imaging features corresponding to various pathologic patterns and may show as a large solitary mass or multifocal nodules of heterogeneous hypoechoic to isoechoic appearances, or a diffuse heterogeneous echogenicity of the whole liver [10]. The echogenicity of the tumor may vary depending on the presence of hemorrhage and necrosis.

CEUS: HAS is characterized predominantly by nodular peripheral enhancement during the arterial phase and portal venous phase, while diffuse chaotic or reticular enhancement could also be seen. In the late phase, HAS typically shows hypoenhancement, possibly with partial rim-like enhancement or isoenhancement, without centripetal filling [10]. In contrast, typical hemangiomas could be accurately diagnosed by CEUS with peripheral nodular enhancement and centripetal filling, which should be differentiated from HAS (Figure 4).

CT: CT scan usually shows hypodense nodules compared to the adjacent liver parenchyma, but may present hyperdense foci representing acute hemorrhage [89]. After intravenous administration of contrast agents, HAS lesions show heterogeneously hypodense FLLs with occasionally peripheral nodular enhancement, central enhancement, or rim-like enhancement on arterial and venous phase images [91]. On delayed images, persistent heterogeneous enhancement is shown with bizarre progressive filling instead of centripetal pattern, possibly due to central fibrosis or necrosis [88].

MRI: HAS lesions generally show as predominantly hypointense on T1-weighted images with hyperintense foci intratumoral hemorrhage, and heterogeneously hyperintense on T2-weighted images with dark septa or fluid levels consistent with hemorrhage [6]. Diffuse heterogeneous signal intensity without a focal mass may be seen less commonly [89]. CEMRI reveals similar contrast-enhanced features to CECT, with markedly heterogeneous enhancement during all phases, with a progressive pattern lacking central filling on delayed phase images [92,93]. The heterogeneous appearances might be associated with the variety of the vascular structures of the tumor. Areas of freely anastomosing channels show rapid enhancement, while areas with large cavernous spaces show low and progressive enhancement [89].

Other imaging techniques: PET/CT scan can demonstrate multiple focal intense accumulations of FDG in liver tumors and may contribute to confirm the presence or absence of distant metastatic diseases at other sites. The uptake of FDG in the tumors might be associated with the overexpression of GLUT-1 and the active proliferation of tumor cells [91].

### 5.3. Management and Prognosis

Pediatric HAS has a limited response to chemotherapy, radiation, and other vascular-targeted agents such as mTOR inhibitors. Complete surgical resection is considered to provide the best hope for disease-free long-term survival, but this may be difficult due to the large size of the tumor [94]. Liver transplantation could be a choice in patients with tumors not amenable to resection, but only a few successful cases have been reported due to high cancer recurrence and post-transplant mortality [94,95]. In addition, transcatheter arterial embolization (TAE) can be used to treat acute tumor rupture, and TACE may represent an alternative therapy for patients with dominant masses [88].

Pediatric HAS has a dismal prognosis regardless of treatment or stage, with a 5-year overall survival less than 30% [57]. Most patients experience a rapid clinical decline and succumb to the disease within 6–12 months [81]. Pediatric patients with IHH that may develop into HAS should undergo serial ultrasound examinations to monitor malignant transformation [2].

## 6. Malignant Rhabdoid Tumor of the Liver (MRTL)

### 6.1. Epidemiology, Clinical Features, and Pathological Features

Malignant rhabdoid tumors were initially described as a highly aggressive variant of Wilms’ tumor; over time, extrarenal locations, including the central nervous system, liver, and other organs, have also been described and reported [96]. Malignant rhabdoid tumor of the liver (MRTL) is a rare and aggressive neoplasm, accounting for approximately 3% of all primary liver malignancies in childhood [97]. The median age at presentation is 8 months, with most patients below 2 years of age [98]. Genetic analyses reveal that MRTL is related to the typical mutations of the SMARCB1 gene [99].

Clinically, patients are often asymptomatic until the lesions become large. Most pediatric patients may present with hepatomegaly, abdominal mass, pain, and a distended abdomen, accompanied by nonspecific systemic symptoms, including fever, anorexia, vomiting, lethargy, malaise, anemia, and weight loss. Spontaneous tumor rupture may occur sometimes, which is more frequent than has been reported for hepatoblastoma or HCC in this age group [100]. Serum AFP level is normal in most cases. About 70% of patients present with metastases on initial presentation, most frequently in the lungs and lymph nodes [98].

Histologically, MRTL is composed of tumor cells with rhabdoid features or basaloid features, staining positively with both epithelial markers and mesenchymal markers [98,101]. Immunohistochemistry shows negative staining for nuclear INI-1 protein, being specific for rhabdoid tumors. Hepatoblastoma with small cell undifferentiated histology can mimic MRTL and show normal AFP levels but lack INI-1 mutations [102]. For infant patients with normal AFP levels at diagnosis, and detailed cytogenetics on imaging results, an immunohistochemical and molecular analysis of INI-1protein might be useful in differential diagnosis between MRTL and hepatoblastoma [103].

### 6.2. Imaging Features

Ultrasound: Grey-scale ultrasound reveals a solitary, large, polycystic, and heterogeneous hyperechoic FLL. Hyperechoic sediment may be identified in the cystic portion [98].

CEUS: There are no CEUS experiences in the literature on MRTL in pediatric patients so far.

CT: MRTL usually demonstrates a large, septate, well-defined, and predominantly hypodense mass on CT scan. CECT may show a predominantly hypodense lesion with discrete peripheral enhancement, consisting of a solid component and a polycystic component. Cystic portions, periphery calcification, and surrounding hematoma due to tumor rupture might be observed. Direct invasion to the adjacent organs, such as the retroperitoneum and right diaphragm, has been reported [104].

MRI: MRTL generally reveals heterogeneous patterns on MRI, with the lesion showing as hypointense on T1-weighted images and hyperintense on T2-weighted images [105].

### 6.3. Management and Prognosis

Multiagent chemotherapy, generally including vincristine, doxorubicin, cyclophosphamide, and other agents, in combination with complete surgical resection remains the primary treatment of MRTL [100]. Jayaram et al. [106] reported a case of successful management of MRTL and suggested that aggressive chemotherapy followed by early liver transplantation should be an option to be considered in unresectable MRTL. Tumor suppressor genes associated with SMARCB1 targeted therapies may have potential applications and offer a greater hope of cure [107,108]. Overall, the outcomes for patients with MRTL are very poor, with a median survival of 2 months [4].

## 7. Conclusions

Special attention should be paid to malignant FLLs diagnosed in pediatric patients. Due to the rarity of these diseases, a higher amount of exact clinical data is required for a deeper comprehension of pediatric malignant FLLs. The application of CEUS in pediatric patients to characterize FLLs remaining indeterminate on conventional B-mode ultrasound may be an effective option in the future, and has great potential to be integrated into imaging algorithms without the risk of exposure to ionizing radiation.

## Figures and Tables

**Figure 1 diagnostics-13-03659-f001:**
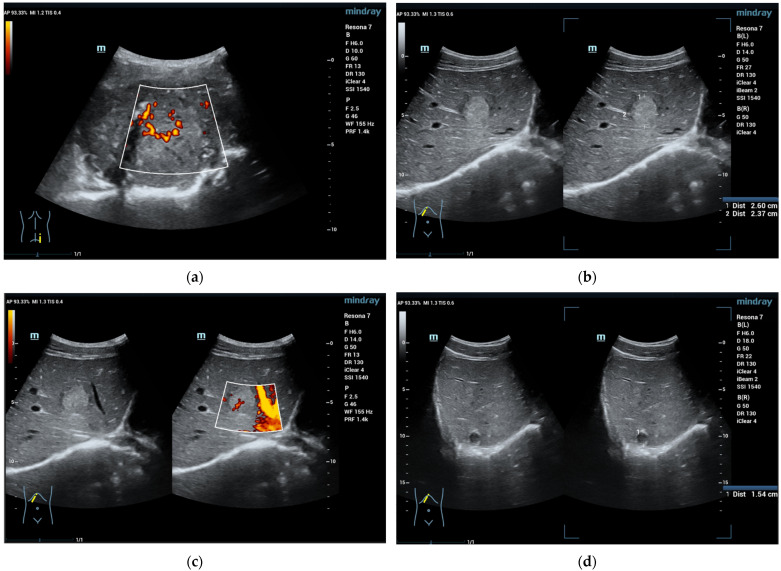
A 16-year-old boy, athletic, soccer player, complaining of pain in the right lumbar–gluteal region for 10 months. MRI showed a large mass in the right iliac bone with protrusion to the muscles. Biopsy result is Ewing sarcoma. This mass is clearly visible on ultrasound (**a**). In liver B mode ultrasound revealed several foci: 26 × 23 mm echogenic with some hypoechoic rim (**b**) and entering vessel on Power Doppler mode (**c**) and 15 mm hypoechoic cystic-like (**d**). CEUS exam showed rapid filling (**e**) with early washout in the middle of the arterial phase (**f**). Washout progresses in the portal venous (**g**) and late venous phases (**h**). Liver metastases of Ewing sarcoma diagnosed. (SC5-1U transducer).

**Figure 2 diagnostics-13-03659-f002:**
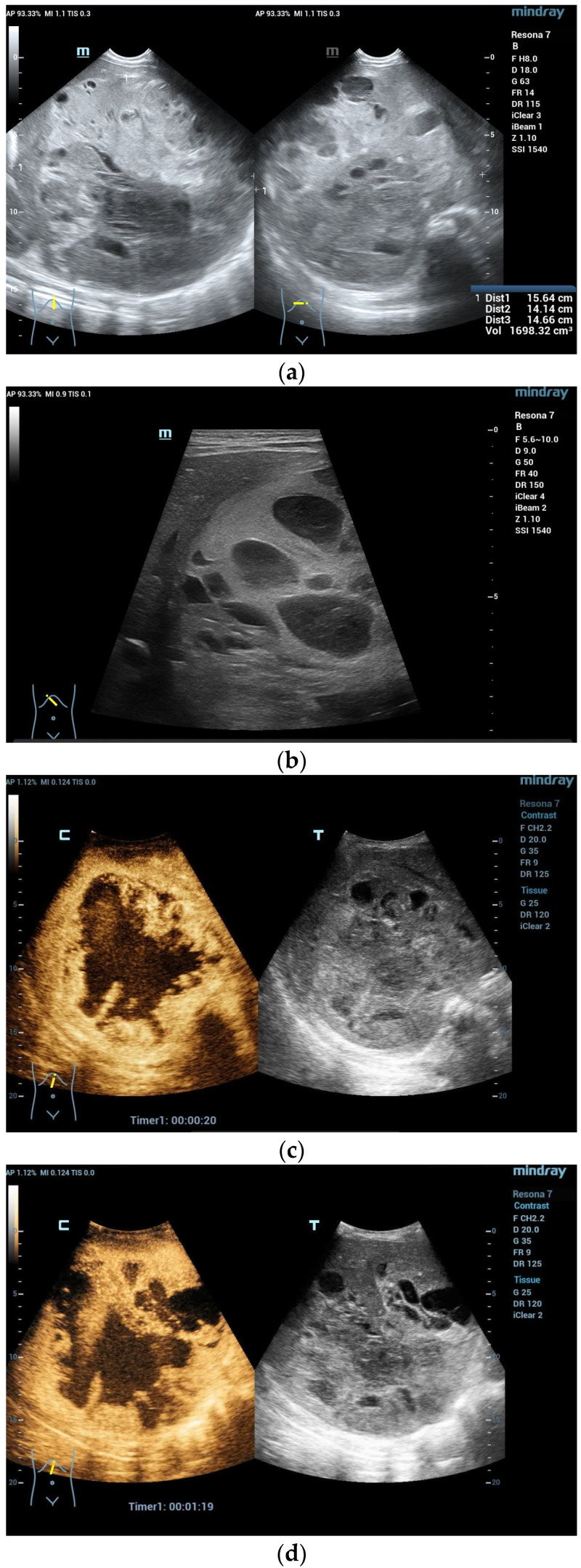
A 5-year-old boy, presenting with a large mass in the left lobe of the liver. B mode shows a heterogenic mass (**a**). High resolution B mode with a linear probe clearly reveals multiple cavities representing hemorrhage and necrosis within the mass (**b**). CEUS in the arterial and early portal venous phases revealed rapid peripheral enhancement with prominent centripetal fill-in and non-enhancing areas (**c**–**e**). Late washout after two minutes could be documented (**f**). Needle biopsy and histological evaluation revealed a highly malignant (G3) liver embryosarcoma.

**Figure 3 diagnostics-13-03659-f003:**
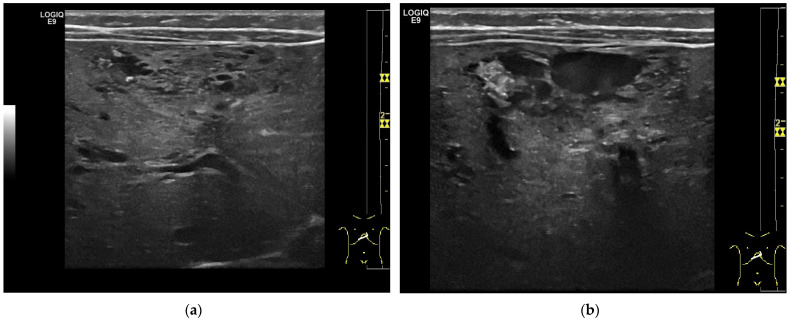
Hemangioendothelioma in a 1-month-old infant. A cystic mass with a maximum diameter of 4 cm in liver segment IV with blurred borders and calcifications with dorsal acoustic shadow (**a**–**e**). Evidence of increased vascularization on color Doppler (**f**). Feeder artery with inflow from the hepatic artery and venous outflow via the markedly dilated left hepatic vein (**g**). Increased flow velocities in the coeliac trunk and hepatic artery (**h**,**i**). Outflowing left hepatic vein with arterialized flow profile and increased flow velocities (**j**).

**Figure 4 diagnostics-13-03659-f004:**
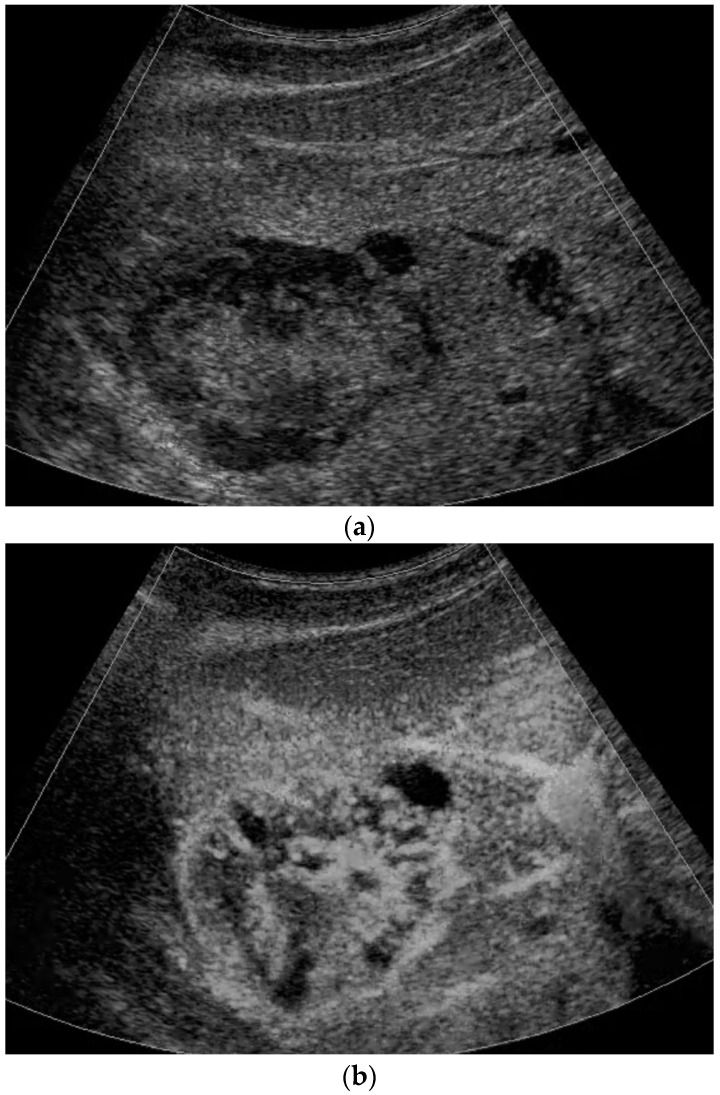
A 17-year-old female with a focal liver lesion in the right liver lobe and a small oval cyst in the surrounding area (**a**). Arterial phase hyperenhancement predominantly in the periphery with a rim and septae-like structures with non-enhancing areas (**b**) and washout are documented (**c**). Needle biopsy with histological evaluation revealed angiosarcoma of the liver.

## Data Availability

Not applicable.

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
