# Peer review of "Review on Pediatric Malignant Focal Liver Lesions with Imaging Evaluation: Part II"

_diagnostics, 2023, doi:10.3390/diagnostics13243659_

Round 1

Reviewer 1 Report

Comments and Suggestions for Authors

Well-written review. Presenting the latest knowledge of the common pediatric malignant FLLs

Knowledge of the pathologic features of these tumors and their imaging appearances are not only helpful for clinical diagnosis, offer an appropriate differential diagnosis but also is useful in the evaluation and follow-up of pre- and post-treatment .

It is worth emphasizing that the authors draws attention to CEUS. Conventional B-mode ultrasound is the well-established first line imaging modality for the assessment of liver pathology in pediatric patients, but US technological advances, the use of contrast agents may improve the ability of ultrasound to distinguish between liver malignant and benign abnormalities and because it can be performed at the same appointment as unenhanced ultrasound, more rapid diagnoses may be possible.

The work is enriched by very good photo documentation

Comments on the Quality of English Language

Minor editing of English language required.

Author Response

We thank the reviewer for the very kind comments. CEUS is also used at point of care and currently also guidelines have been published for “CEUS acute” [Michels G et alii, Standardized contrast-enhanced ultrasound (CEUS) in clinical acute and emergency medicine and critical care (CEUS Acute): Consensus statement of DGIIN, DIVI, DGINA, DGAI, DGK, OGUM, SGUM and DEGUM]. Med Klin Intensivmed Notfmed 2022;117:1-23 and Anaesthesist 2022; 71: 307-310].

Reviewer 2 Report

Comments and Suggestions for Authors

The authors' effort to produce a review manuscript on pediatric malignant FLL with imaging evaluation is commendable, as radiologists should be familiar with the main imaging findings, clinical features and management (both diagnostic and therapeutic) of children with such overall rare diseases.

However, also in consideration of the emphasis on imaging evaluation, I would expect MANY more figures (ideally from multiple imaging modalities) illustrating imaging findings from real patient cases for each illustrated FLL. Actually only two figures containing ultrasound and Doppler images have been provided. This is a serious flaw of the manuscript that is to be addressed.

Moreover, it would be strongly advisable to not only describe the imaging findings of FLL, but also the rationale for the use of the various imaging modalities for the diagnostic and therapeutic workup of pediatric patients with FLL. The correlation between imaging and pathological findings should also be discussed systematically wherever possible.

Comments on the Quality of English Language

Moderate English language editing would be needed.

Author Response

Point 1: The authors' effort to produce a review manuscript on pediatric malignant FLL with imaging evaluation is commendable, as radiologists should be familiar with the main imaging findings, clinical features and management (both diagnostic and therapeutic) of children with such overall rare diseases. However, also in consideration of the emphasis on imaging evaluation, I would expect MANY more figures (ideally from multiple imaging modalities) illustrating imaging findings from real patient cases for each illustrated FLL. Actually only two figures containing ultrasound and Doppler images have been provided. This is a serious flaw of the manuscript that is to be addressed.

Response: Thanks for your helpful advices. This review summarized the imaging features of malignant FLLs in pediatric patients in 18 images and more images are prepared for part I. We displayed only the ultrasound images since the report is focussed on ultrasound. The comparison of image methods will be illustrated in future research. The amount of images are limited by the journal.

Point 2: Moreover, it would be strongly advisable to not only describe the imaging findings of FLL, but also the rationale for the use of the various imaging modalities for the diagnostic and therapeutic workup of pediatric patients with FLL. The correlation between imaging and pathological findings should also be discussed systematically wherever possible.

Response:

We described the principles of ultrasound technologies for the diagnostic and therapeutic workup of pediatric patients with FLL in the Part I of the review. We focus on the imaging features of FLLs and discuss the correlation between imaging and pathological findings according to the current knowledge. The special issue does not allow additional text and images (so far 18 figures), therefore, the article was divided into two.

We also made a few minor corrections in word track modus and added: “HEHE must be distinguished from the infantile hemangioendithelioma of the liver, which occurs in the newborn and is more likely to be classified as a congenital hemangioma (Figure 2).”

Round 2

Reviewer 2 Report

Comments and Suggestions for Authors

Only few minor changes have been made to the first version of the manuscript, so the main issues highlighted in my revision have remained substantially unresolved.

Comments on the Quality of English Language

Moderate editing of English language would be required.

Author Response

Point 1: The authors' effort to produce a review manuscript on pediatric malignant FLL with imaging evaluation is commendable, as radiologists should be familiar with the main imaging findings, clinical features and management (both diagnostic and therapeutic) of children with such overall rare diseases. However, also in consideration of the emphasis on imaging evaluation, I would expect MANY more figures (ideally from multiple imaging modalities) illustrating imaging findings from real patient cases for each illustrated FLL. Actually, only two figures containing ultrasound and Doppler images have been provided. This is a serious flaw of the manuscript that is to be addressed.

Response: Thanks for your helpful advices. This review summarized the imaging features of malignant FLLs in pediatric patients in 18 images and more images are prepared for part I [DOI: 10.3390/diagnostics13233568]. We displayed only the ultrasound images since the report is focussed on ultrasound and this is still true for the 2nd revision. The comparison of image methods will be illustrated in future research. Additional figures have been added, see under comment by the editor.

Point 2: Moreover, it would be strongly advisable to not only describe the imaging findings of FLL, but also the rationale for the use of the various imaging modalities for the diagnostic and therapeutic workup of pediatric patients with FLL. The correlation between imaging and pathological findings should also be discussed systematically wherever possible.

Response:

We described the principles of ultrasound technologies for the diagnostic and therapeutic workup of pediatric patients with FLL in the Part I of the review [DOI: 10.3390/diagnostics13233568, published on 2023 29th of November].

“Generally, ultrasound is used as an initial screening technique to detect the lesion and rule out any other abnormalities, followed by computed tomography (CT) and/or magnetic resonance imaging (MRI) to complete characterization and staging of tumors. The use of CT or MRI before surgery is mandatory. All mentioned imaging techniques have their advantages and disadvantages.”

Since the review did not allow additional text and images, therefore, the article was divided into two. We focus on the imaging features of FLLs and discuss the correlation between imaging and pathological findings according to the current knowledge.

“Because hepatic sarcomas are extremely rare, only small case studies and case reports have been published. Hepatic sarcomas likewise other FLL can be classified in different types according to their growth pattern and degenerative histological features: a large solitary mass, a mixed pattern with a dominant mass associated with satellite nodules, multiple nodules and the rare form of diffuse infiltrating micronodular tumor and metastases [Trojan J, Hammerstingl R, Engels K, Schneider AR, Zeuzem S, Dietrich CF. Contrast-enhanced ultrasound in the diagnosis of malignant mesenchymal liver tumors. J Clin Ultrasound 2010;38:227-231]. Pathologically, angiosarcomas are characterized by areas of hemorrhage and necrosis [Kim KA, Kim KW, Park SH, Jang SJ, Park MS, Kim PN, Lee MG, et al. Unusual mesenchymal liver tumors in adults: radiologic-pathologic correlation. AJR Am J Roentgenol 2006;187:W481-489], therefore, imaging features of hepatic angiosarcoma show typically bizarre heterogenous lesions, which might be sometimes misdiagnosed as atypical gigantic hemangioma with non-enhancing areas [Dietrich CF, Mertens JC, Braden B, Schuessler G, Ott M, Ignee A. Contrast-enhanced ultrasound of histologically proven liver hemangiomas. Hepatology 2007;45:1139-1145]. Coarse nodular enhancement with or without centripetal filling combined with the presence of a rim and reticular ‘‘septal-like’’ chaotic arterial enhancement and wash out should prompt further diagnostic workup including a needle biopsy. It is of importance that wash out might be late, sometimes after two and up to five minutes.”

We also made a few minor corrections in word track modus and added: “HEHE must be distinguished from the infantile hemangioendothelioma of the liver, which occurs in the newborn and is more likely to be classified as a congenital hemangioma (Figure 2).”

Round 3

Reviewer 2 Report

Comments and Suggestions for Authors

This revised version is definitely better than the previous ones. Thank you.

Comments on the Quality of English Language

A moderate English language revision is still needed.